# An Overview of Orthoptera Mass Occurrences in Croatia from 1900 to 2023

**DOI:** 10.3390/insects15020082

**Published:** 2024-01-23

**Authors:** Niko Kasalo, Nikola Tvrtković, Domagoj Bogić, Bože Kokan, Marijana Vuković, Mladen Kučinić, Josip Skejo

**Affiliations:** 1Independent Researcher, Matice Hrvatske 11, BA-80101 Livno, Bosnia and Herzegovina; 2Independent Researcher, Alagovićeva 21, HR-10000 Zagreb, Croatia; nikolatvrtkovic71@gmail.com; 3Division of Zoology, Evolution Lab, Department of Biology, Faculty of Science, University of Zagreb, Rooseveltov trg 6, HR-10000 Zagreb, Croatia; mascer2@gmail.com (D.B.); mladen.kucinic@biol.pmf.hr (M.K.); 4Natural History Museum Split, Kolombatovićevo Šetalište 2, HR-21000 Split, Croatia; boze@prirodoslovni.hr; 5Croatian Natural History Museum, Demetrova 1, HR-10000 Zagreb, Croatia; mvukovic@hpm.hr

**Keywords:** Mediterranean region, locusts, grasshoppers, katydids, mole crickets, pest species, abundance, outbreaks (hrv. *najezda*)

## Abstract

**Simple Summary:**

Grasshopper and cricket (Orthoptera) mass occurrences represent an unusually large increase in populations of certain species. Grasshopper species known as locusts are well known to cause severe damage to many crops. The severity of each outbreak depends on the species and on a wide range of ecological factors, including temperature and rainfall regime. In Croatia, a small country in SE Europe, there has never been systematic research on these Orthoptera species, and this phenomenon is regarded as a rare event despite several huge outbreaks being reported in the past. For the first time, all known mass occurrences of grasshoppers, locusts, crickets, bush-crickets, and mole crickets (of the order Orthoptera) from Croatia are summarized. The study represents a historical overview of all Orthoptera mass occurrences known based on the literature but also provides many new records. It seems that Orthoptera mass occurrences are not rare in the country at all, especially in the Mediterranean part. The present situation suggests that thorough research into locusts as agricultural pests in Croatia should be conducted to determine the potential threat to the local economy and with special emphasis on the Adriatic region. The trends of climate change in the Mediterranean imply a rise in locust destructiveness risk in the near future and invites further research.

**Abstract:**

During the last century, well-known locust species, such as *Calliptamus italicus* and *Dociostaurus maroccanus*, have produced outbreaks of varying degrees in the Balkans. The literature data on outbreaks in the region are scarce, and Croatia is not an exception. This study summarized the data on 23 Orthoptera mass occurrences in Croatia from 1900 to 2023 from 28 localities, representing 12 species. This is a low level of outbreak activity compared with other locust and pest grasshopper species in other parts of the world. The species with the most reporting is *C. italicus* with altogether six mass occurrences, while second is *Barbitistes ocskayi* and *Miramella irena* with three records, and in the third, place *D. maroccanus* and *Gryllotalpa* sp., each with two mass occurrences having been reported. One of the most notable swarms is that of *Anacridium aegyptium* which occurred around Šibenik in 1998, and this paper provides the first account of it, 25 years after it took place. The most recent outbreaks took place in 2022, and the most notable one was that of *D. maroccanus* swarm in Štikovo. The 2022 and 2023 reports were brief and muted, despite the affected agriculturists claiming significant damages.

## 1. Introduction

Mass outbreaks of Orthoptera are a well-known phenomenon, both biologically and economically [1,2]. Certain species of grasshoppers belonging to the family Acrididae (Orthoptera: Caelifera) even exhibit density-dependent phase polyphenism in many aspects of their morphology, development, biochemistry, and ecology [3,4,5,6,7]. Swarming grasshopper species have been well known to people since ancient times, and their outbreaks were already reported in the Bible and Qur’an [8]. Uvarov [1,9] introduced the terms “solitarious” and “gregarious” to denote the solitary and the swarming phases, respectively, but intermediate forms are also known [5]. The induction of a phase change is also continuous, not specific to a single life stage, and reversible [10]. There are many factors regulating these phase changes but the essential trigger for inducing the gregarious phase is a large population density [5,10,11,12].

Although locust species represent a small fraction of grasshopper species [13], they should be monitored as their large population densities and the accompanying voracity make them economically important pests that can damage crops in large areas [14,15]. Different species have different biologies and produce swarms of varying severities. *Dociostaurus maroccanus* (Thunberg, 1815) [16] is a species that produced severe multi-year plagues in the past [2,14,17,18], but today, severe outbreaks are reported only from Sardinia [19], and herewith from Croatia. On the other hand, *Calliptamus italicus* (Linnaeus, 1758) [20] is an example of a species that has on several occasions destroyed millions of hectares of land and which remains a very present agronomic threat [15].

The Balkan peninsula and the surrounding area of the Pannonian lowland seem to have been plagued by locust swarms for hundreds of years [21,22,23], with the Moroccan Locust—*Dociostaurus maroccanus,* and the Italian Locust—*Calliptamus italicus,* being especially destructive. Gradojević [21,22,23] lists Gevgelija (North Macedonia) and Kikinda (Vojvodina, Serbia) as known gregarious zones of the Moroccan locust, and Podgorica (Montenegro) and Stolac (Herzegovina) as gregarious zones of the Italian locust. Of the more recent breakouts, Gradojević [22,23] mentions the large flying swarms of the Moroccan locust in northeast Yugoslavia that caused severe agricultural damage between 1931 and 1933. In the end, the problem was solved by the trampling of the nymphs. Despite the mass Orthoptera outbreaks being known to humans since antiquity, only sporadic data about the swarms and mass appearances in Croatia exist. From the scientific literature standpoint of view, there are only five published records of Orthoptera mass occurrences in the country [24,25,26,27,28]. The recent increase in locust outbreaks in Europe [19] and abroad [29] provides further alarm and reason to summarize Orthoptera outbreaks in our own country as well.

The aim of this study is to summarize data on all the Orthoptera mass occurrences in Croatia, from 1905 to 2023, by presenting an overview of the 11 literature records, and by reporting data on 12 mass occurrences for the first time. The aim regarding the damage-causing species is to establish the precedence of reporting the outbreaks, which should allow for a better assessment of the true dangers posed by them, with a special focus on locusts. The aim regarding the non-damaging species is to call attention to fluctuations in their abundances, which might prove important in elucidating their population dynamics and might be of special importance in the monitoring of the endemic species covered in this study.

## 2. Materials and Methods

A literature review was performed to find the oldest reported records of the Orthoptera mass occurrences in Croatia [24,25,26,27,28].

The data on *Calliptamus italicus* (Linnaeus, 1758) [20]; *Miramella irena* (Fruhstorfer, 1921) [30]; *Psorodonotus illyricus* Ebner, 1923 [31]; *Polysarcus denticauda* (Charpentier, 1825) [32]; *Barbitistes ocskayi* Charpentier in von Ocskay et al., 1850 [33]; and *B. yersini* Brunner von Wattenwyl, 1878 [34] mass occurrences were noted by the authors from 2015 to 2023 during the Croatian Orthoptera fieldwork surveys. A massive occurrence of *Calliptamus italicus* in the Boljunsko polje field was noted by Darko Šimetić in the summer of 2019 from Kurelovići, who contacted Nikola Tvrtković. After Mr. Šimetić’s appeal, the fieldwork was conducted.

The information on the *Prionotropis hystrix* (Germar, 1817) [35] Troglav outbreak in 2012 was kindly provided to us by Lucija Ivić, a colleague theriologist and ichthyologist who happened to visit the mountain during the outbreak.

The information on the outbreak of *D. maroccanus* (Thunberg, 1815) [16] in Štikovo was reported to the authors on 7 June 2022 by the local populace. One of the authors (BK) visited the locality and sampled the locusts by use of a sweeping net. The collected specimens were preserved in ethanol and were later used for identification and obtaining measurements. The tegmen (also known as elytron) length to hind femur length ratio, i.e., the E/F ratio, was calculated for some specimens from this outbreak in order to check whether they presented gregarious morphology.

The information on the outbreak of *C. italicus* on the island of Brač was relayed to the authors by a news reporter on 14 July 2022. The identification of the species was carried out based on photographs provided by the reporter. The information on the mass appearance of *Gryllotalpa* in Sinj was reported by a single news outlet [36]. None of the authors personally visited the latter two localities, so the gathered information is treated with greater scrutiny. The information pertaining to the mass appearance of the *Barbitistes* Charpentier, 1825 species was gathered during one of the authors’ fieldwork trips. The identification of specimens was carried out using the key by Harz [37]. The collected adult specimens from Štikovo were photographed next to a ruler and measured using ImageJ 1.53t software, while the photograph of a single male specimen from Brač was used to determine the relative measurements of the specimen. The length of the femora and the elytra were measured in order to calculate the ratio between these measures, which is often used as a surrogate measure to express if a locust is in a solitarious or gregarious phase [38]. There is video documentation of the Štikovo Swarm on YouTube “https://youtu.be/Sy_jQ0ViAmw (accessed on 12 November 2023)”, and “https://youtu.be/3ZkLwm-QoEY (accessed on 12 November 2023)”.

The information on the 2023 mass occurrence of *Decticus albifrons* (Fabricius, 1775) [39] in Kaštela, Split, and Omiš was reported by several news outlets [40,41]. Furthermore, one of the authors (DB) observed a part of this mass occurrence and recorded a large number of males singing.

Croatian media outlets [36,42,43,44,45] were consulted in order to see whether there were swarms (hr. *najezda, najezda skakavaca*) that were not reported by scientists but were noted by local people.

Finally, interviews with local people were conducted in order to obtain data on several outbreaks. Especially notable is the reconstruction of the *Anacridium aegyptium* (Linnaeus, 1764) [46] outbreak that occurred in 1998 around Šibenik and Primošten and was hitherto never reported. JS, who spent his childhood in the area, remembers the swarm from when he was 6 years old, so the interviews were conducted with professors from the primary school (Nadica Paškvalin), as well as with local people (Iva Kundid) who remember the massive outbreak in summer of 1998. These accounts were used to establish key points of the outbreak. The report relays information as it was given to us.

All the localities mentioned in this study were approximately georeferenced to two decimal places only and are shown in Table 1. The taxonomic account of all the species follows the Orthoptera Species File [47]. The data on all the known Orthoptera mass occurrences in Croatia are summarized chronologically in the Results section.

## 3. Results

The taxonomic account of all the species is shown in Table 2, while the data on all the known Orthoptera mass occurrences in Croatia are summarized chronologically in Table 3.

### 3.1. A General Overview of the Mass Occurrences

There were altogether 23 reports on mass Orthoptera occurrences in Croatia, of which 11 are known from the literature [24,25,26,27,28], while 12 outbreaks are being reported for the first time: (1) *Anacridium aegyptium* 1998 around Šibenik; (2) *Miramella irena* 2009 in Velebit Mt.; (3) *Prionotropis hystrix* 2012 in Troglav Mt.; (4) *Psorodonotus illyricus* 2013 in Poštak Mt.; (5) *Polysarcus denticauda* 2014 in Podbitoraj; (6) *Barbitistes ocskayi* and *B. yersini* 2014 on Krk Island; (7) *Calliptamus italicus* in 2019 Boljunsko polje; (8) *Miramella irena* 2021 in Janjče; (9) *Dociostaurus maroccanus* 2022 In Štikovo, Svilaja Mt.; (10) *Calliptamus italicus* 2022 on Brač island; (11) *Gryllotalpa* sp. 2022 in Sinj, NK Junak field; and (12) *Decticus albifrons* 2023 in Kaštela, Split, and Omiš.

Taxonomically, the reports belong to 12 Orthoptera species, of which 11 are known at the species level (Table 2), while for the 1920s Krk island outbreak, it is not known which Gryllidae species caused the outbreak [25]. The species belong to the families Tettigoniidae (five species), Gryllotalpidae (one species not identified at the species level), Acrididae (four species), and Pamphagidae (one species).

Sinj is the city with the highest number of mass Orthoptera appearances, with altogether three from 1923 to 2022; however, this is a low frequency and likely of novelty value only. Geographically, Krk island was the best-studied island concerning mass occurrences, with 6 out of 28 localities summarized in this study belonging to this island (Table 1). This is mostly due to the 1905–1906 *Barbitistes ocskayi* outbreak, which was thoroughly studied in the local forestry [24,25] (Table 3). It should be noted that other outbreaks could have occurred, but were not reported.

There are “only” five Orthoptera outbreaks in Croatia hitherto reported which are confirmed to have caused severe damage: (1) the 1905–1906 *Barbitistes ocskayi* outbreak on Krk island which caused damage to 240 hectares of young maple trees [24,25]; (2) the 1923 *Dociostaurus maroccanus* outbreak in Sinj which caused damage to crops [26]; (3) the 1946–1947 *Calliptamus italicus* outbreak in Mirna Valley (Istria) which damaged 600 hectares of cultivated land [27]; (4) the 1998 *Anacridium aegyptium* outbreak in the surroundings of Šibenik which also caused large damage to crops; (5) and the 2022 *Dociostaurus maroccanus* outbreak in Štikovo which caused significant damage to the crops in the village.

The species with the most outbreaks in Croatia is definitely *Calliptamus italicus*, with altogether six mass occurrences, followed by *Miramella irena* with three records. *Dociostaurus maroccanus*, *Decticus albifrons, Gryllotalpa* sp. and *Barbitistes ocskayi* are each represented with two mass occurrences (Table 3). The outbreak of *Anacridium aegyptium* was reported only once in Croatia, just like the outbreaks of *Prionotropis hystrix*, *Psorodonotus illyricus*, *Polysarcus denticauda*, and *Barbitistes yersini*, but unlike them, it caused widespread agricultural damage.

It is important to note that there were probably many more natural Orthoptera mass occurrences in the wild, similar to the reported outbreaks of *Prionotropis hystrix* or *Psorodonotus illyricus*, which are Dinaric endemic species [13]. However, due to the lack of systematic orthopterological research in the country prior to the 2010s, these were rarely noted.

### 3.2. A Brief Chronological Overview of the Outbreaks

#### 3.2.1. Earliest Reported Mass Occurrences from Croatia, 1900–1923

For two locust outbreaks (1900 in Pag Island and 1913 around Vrlika), it is not known which species caused them [26], so we cannot add anything new towards their understanding. At least partly, these outbreaks may be attributed to *C. italicus,* as Novak [26] claims.

The earliest reported Orthoptera outbreak in Croatia for which the species is known was that of *Barbitistes ocskayi* on Krk Island in 1905 [24]. Langhoffer [25] lists this species as the only known Orthoptera pest on the Adriatic coast. This species caused severe damage to 240 hectares of young *Acer monspessulanus* bushes in several localities (Table 3 and Figure 1) and also a whole range of their host plants, including *Fraxinus*, *Pistacia*, *Prunus mahaleb*, and *Paliurus* [24,25]. Confusion occurred in the earliest records because certain authors reported *Barbitistes ocskayi* swarming together with *Dinarippiger discoidalis* and *Eupholidoptera schmidti* [49], but in fact, these species were reported to feed on swarming *Barbitistes ocskayi* [24,25]. It was even mentioned that the local name for this pest is “kršuljka” [25,49]. The outbreak on Krk did not happen in a single year, but it continued in 1906. This outbreak was notable, as it was reported in three publications [24,25,49].

Besides the Pag 1900 outbreak and the Vrlika 1913 outbreak of unknown identity, Novak [26] reported more locust mass occurrences. An outbreak attributed to *C. italicus* happened on Vis Island in 1922, while in 1923 in Sinj, there was an outbreak caused by *Dociostaurus maroccanus*. The populace fought against the locusts by destroying eggs and by forcing the nymphs to a single area which was then burned [26]. Besides the locust outbreaks, Novak (1928) reported a large outbreak of bush crickets, *Decticus albifrons,* in 1923 in Sinj.

Langhoffer [25] briefly mentioned that an unidentified species of a cricket belonging to the genus *Gryllus* Linnaeus, 1758 [20] caused damage in a garden in Baška on Krk Island and that a species of the genus *Gryllotalpa* Latreille, 1802 [50] caused damage to multiple sapling nurseries in Benkovac.

#### 3.2.2. Mass Occurrences Reported in Croatia from 1946 to 1987

Kovačević [27] confirms that *C. italicus* is a known pest in North Macedonia, Montenegro, and Croatia (namely Dalmatia and Istria). He reports the destruction of 600 hectares of cultivated land in the valley of Mirna in Istria in 1946 and 1947 by *C. italicus*. He also notes that *D. maroccanus* is especially problematic in North Macedonia, Montenegro, and Herzegovina [27]. Herzegovina is especially interesting as it neighbors Dalmatia and has a similar climate. Mikšić [51] briefly summarizes some more Moroccan locust outbreaks in SE Bosnia and Herzegovina that happened in 1922 and between 1946 and 1947. She also reports an outbreak that happened in the years 1975 and 1976 around Mostar and in Popovo polje. Although the report is non-systematic and at times even convoluted, it can be gleaned from it that during those years, *D. maroccanus* appeared in unusually large numbers, represented mainly by nymphs. The gathered adults were determined to be in a transient form between gregarious and solitarious. The outbreak was locally contained, and no damage to crops was observed, potentially because the crops were preventively treated with insecticides [51]. Another recent breakout of the Moroccan locust occurred in a relative vicinity, in Albania in 2014 [52] but was not described in detail. The Moroccan locust thus seems to be a relatively common threat in the areas with Mediterranean influence, but detailed reports are lacking.

Pavićević & Karaman [28] reported a *Miramella irena* outbreak in 1987 in the forestry near Osijek. The specimens were provided to Pavićević and Karaman by a Belgrade Faculty of Forestry professor Lj. Mihajlović. The specimens were collected on 15.5.1987. by ing. Lovas who observed them damaging young tree sprouts. There are no specifics about the forestry in which the outbreak happened. This ambiguity leaves several possible candidates for the true locality of the 1987 *Miramella irena* outbreak—Valpovo Forestry, Darda Forestry, and Tikveš-Bilje Forestry.

Kovačević [27] notes that *Anacridium aegyptium* is a minor pest affecting tobacco, and that it only ever appears in small numbers. In general, *A. aegyptium* produces very sporadic and localized outbreaks that damage vineyards [53,54]. Here, we report a nearly forgotten outbreak of this species—the first for the Balkans.

#### 3.2.3. Reconstructing a 25-Year-Old Šibenik *Anacridium aegyptium* Swarm from Multiple Personal Accounts

While we were gathering data for this paper, Josip Skejo remembered a scene from his childhood. Many years ago, during the summer festivities known as “Primoštenske užance” in Primošten town, there was a large grasshopper presence. He vividly remembers himself and other children collecting the dead grasshoppers in their shirts and feeding them to cats. Another highly personal memory of the swarm was revealed in conversation with Nadica Paškvalin, who had a hip surgery that summer. She remembers sitting on her porch when grasshoppers jumped at her feet. Surprised and frightened, she ran into her house, forgetting the crutches she had been using following her surgery.

The year the swarm happened was narrowed down according to several accounts. Aleksandra Laćak expressed certainty that it happened after 1997. Nadica Paškvalin and her husband Frane Paškvalin, as well as Iva Kundid, claim that the year was 1998, while Dubravka Skejo is certain that it happened before the birth of her daughter, which was in 1999. Mirjana Lovrić believes that 1998 was not the year it happened, but admits that she does not remember the year very well. Thus, 1998 is established as the most likely year when the swarm appeared. The areas that were afflicted are Materize on the outskirts of Šibenik (Dubravka Skejo pers comm.), Donje polje near Šibenik (Dane Kundid), the Primošten outskirts (Josip Skejo), and the area between the two towns (Nadica Paškvalin pers. comm.). In essence, the entire wider area was hit by the outbreak. There were almost no grasshoppers inside the towns, only on the outskirts, in the villages, and on croplands (Aleksandra Laćak). Grapevine and olive plantations suffered heavy damage (Dane Kundid, Nadica Paškvalin, Frane Paškvalin pers. comm.).

There are no recorded attempts at controlling the outbreak, and it was evidently constrained only to that one season, never to appear again. However, Dane Kundid remembers that when a lot of grasshoppers appeared during the 1950s and 1960s, they would release turkeys to eat them. As Josip Skejo remembers, the species in question was *Anacridium aegyptium*, and the descriptions by other observers leave little doubt that this large and unique grasshopper is to blame. As a side note, Iva Kundid reports that in recent years, marked by dry hot summers and warm winters, grasshoppers have become more common and are thus becoming a more significant pest. There is no information about the level of damage sustained by crops in recent years, nor is it known which species cause it. This lack of information is a systemic issue—the local institutes of public health seemingly track larger orthopteran outbreaks, but they do not distinguish caeliferan and ensiferan outbreaks, and no concrete information is publicly available [41].

#### 3.2.4. Recent Mass Occurrences in Croatia from 2009 to 2023

Because of the start of systematic orthopterological research in Croatia in 2010, there are many records of Orthoptera mass occurrences from 2009 on (11 out of 23 reports from this study); some are reported only briefly, while others are reported more systematically.

In July 2009, N. Tvrtković, M. Vuković, and I. Mihoci observed a mass occurrence of *Miramella irena* on Velebit Mt. in Bužim (Table 3). Interestingly, N. Tvrtković and M. Vuković observed *Miramella irena* again in 2021 in Odmorište Janjče station in very high abundance (Figure 2). Besides the 1987 [28] report of the *Miramella irena* mass occurrence, no other data are available. With the new data, it may be suspected that this species sporadically appears in high abundances, which could lead to the damage of certain plant cultures if the outbreak occurs in a cultivated area. The population dynamics of *M. irena* remain unknown, which is a research gap that should be addressed in the future.

In the summer of 2012, during a biology students’ expedition—BIUS to Dinara and Troglav Mts.—Lucija Ivić and her colleagues noted the mass occurrence of *Prionotropis hystrix*, a Dinaric karst endemic species, around a Vrdovo mountain house on Troglav Mt. This is an interesting observation because it is not the only example of a local endemic mass occurrence. In June 2013, N. Tvrtković and M. Vuković observed the mass occurrence of *Psorodonotus illyricus* in Poštak Mt., Ljubina Poljana, while in August 2014, N. Tvrtković observed an outbreak of *Polysarcus denticauda* in Podbitoraj.

On 7 and 8 June 2015, a mass appearance of *Barbitistes ocskayi* and *B. yersini* was observed near Ponikve lake on Krk Island by J. Skejo and C. Roesti. There are no estimates on the population densities and no behavioral notes, but it was recorded that the darker color forms were dominant in the swarming populations of both species.

The most notable recent locust swarm was that of *Dociostaurus maroccanus* in Štikovo. The collected material from Štikovo comprised almost exclusively of fourth- and fifth-instar nymphs. Only two adult males were collected, and their E/F ratios were 1.66 and 1.69. Field observations confirmed a large number of individuals of *D. maroccanus* in the surveyed area. An approximation of the population density cannot be made as the fieldwork was brief. The locals reported that the swarms had been appearing for four years with increasing severity, but there are no official reports from the last three years. Although damage was confirmed first-hand, no official damage assessment was carried out. The outbreak happened in early June of 2022 and was reported on by Croatian news outlets [42,43,45]. A photograph of a part of the swarm is shown in Figure 3.

In 2023, the Večernji list [55] published a short video of a supposed locust swarm on Brač island. This video is embedded in an article about a *Decticus albifrons* mass occurrence and provides no detail about the precise location or time. The specimens visible in the video appear to be *D. moroccanus*, but no experts were notified about this swarm, and thus, no fieldwork was conducted.

The outbreak of *C. italicus* on Brač was first reported from the village of Selca and followed up by several sources. The interviewed local reported that the locusts have spread to the rest of the island, but there are no first-hand reports to confirm this. The interviewee and a local photographer claim severe damage to their crops and unsuccessful attempts to establish communication with the local authorities regarding this issue. Unfortunately, no conclusive information on this outbreak could be gathered, so its real extent is impossible to ascertain. The E/F ratio of a single photographed male is 1.27. The outbreak happened around the middle of July 2022 and was reported on by a single Croatian news outlet [44]. No photographs of the swarm were taken, only photographs of individual specimens.

The mass appearance of *Gryllotalpa* sp. [36] in Sinj in 2022 could not have been followed up on. After a period of abundant rainfall combined with pesticide use, a large number of *Gryllotalpa* sp. specimens appeared on the surface of a football field in Sinj. There is no estimate of the population density, and it is unclear in what numbers the species normally appears in the area. A photograph of a part of the reported mass appearance is shown in Figure 4.

Lastly, in 2023 there was a massive upsurge in the population of *Decticus albifrons* in the Dalmatian region, most notably in Kaštela, Split, and Omiš. Considering the species’ mostly carnivorous diet, no damages were reported, but due to its large size, many independent reports of sightings of a large number of specimens are available.

## 4. Discussion

### 4.1. Sporadic (or Severely Overlooked) Ensiferan Mass Occurrences

Crickets, bush crickets, and mole-crickets do not produce as significant nor as frequent outbreaks as grasshoppers do [56]. A rare exception are the mole crickets, *Gryllotalpa* spp., which are known as a pest of potatoes, carrots, and other crops [57]. Strangely, outbreaks of *Gryllotalpa gryllotalpa* are rarely reported, but they are nevertheless subjected to pest control, making their numbers dwindle [57]. The mass appearance of *Gryllotalpa* sp. in Sinj in the NK Junak football field occurred after heavy rain combined with pesticide use, so it cannot be considered a true outbreak.

*Barbitistes ocskayi* is reported to cause leaf damage [25] that can be significant, but a severe outbreak was recorded only once. This aspect of the species’ biology is a potential target for future research. The mass occurrences of *Psorodonotus illyricus* and *Polysarcus denticauda* are interesting; they were short-lasting as birds and other animals quickly ate many bush-crickets. This may be a normal part of the species’ population dynamic [58], which is understudied in Ensifera in general, making it important to report such occurrences [56,59]. Similarly, the 2023 *Decticus albifrons* population was short-lived and caused no economic harm. In any case, the ensiferan outbreaks have not been observed to pose a risk for agriculture in Croatia. Most of the recorded mass occurrences in Croatia were reported from areas far away from densely populated centers, indicating that there is a possibility that a lot of these population booms pass unnoticed. The *D. albifrons* mass occurrence occurred in densely populated areas and was quickly followed by sensationalistic media reports.

### 4.2. Non-Locust Caeliferan Mass Occurrences

Several non-locust caeliferan species were observed in high abundance, namely *Miramella irena*, *Prionotropis hystrix*, and *Anacridium aegyptium*. Although *A. aegyptium* is known as the Egyptian locust and although it exhibits density-dependent color polyphenism, it notably does not form migrating swarms like true locusts [3] and is thus excluded from the discussion on locusts.

No records of damaging outbreaks of *M. irena* exist, but a related species, *M. alpina*, is known to occasionally occur in high abundance [60], sometimes causing the extensive defoliation of forest vegetation [61]. The reason for these events remains to be studied. In general, the abundance of caeliferan species are known to be influenced by habitat composition and complexity [62,63], but they also vary in time, with cyclical population booms [60]. Although population dynamics in grasshoppers are still not well understood in the long term, it is clear that some species are more prone to outbreaks than others [60]. There are no recorded mass occurrences of *P. hystrix*, a species that has been assessed by the IUCN as vulnerable due to shrinking populations and areas of occupancy [13]. Data on this species are lacking, so how common its mass occurrences are and what exactly causes them should be researched.

As exemplified by our reconstruction of the Šibenik 1998 *A. aegyptium* outbreak, there is great value in engaging with the local community while conducting research. The foremost scientific benefit is in gathering a larger number of reports than is usually possible, and those reports can be highly accurate when they are evaluated as a whole [64]. Alongside this, local communities provide insight into how they perceive biological problems at the everyday, practical level. Our brief report maintains a slight anthropological tone, noting the anchoring memories that guided people’s recollections, in an effort to provide a snapshot into that brief moment 25 years ago. The fact that such a well-known local story was never properly reported follows the theme of this paper—a lot of information slips away when no one is listening.

*Anacridium aegyptium* occurs in high abundance only rarely, and the causes of this phenomenon are not understood [53,54]. The fact that it is a pest of grapevines is corroborated by our findings, but we do not have any conclusions regarding why it could be like this. It should be kept in mind, nevertheless, that *A. aegyptium* is expanding its distribution due to global warming [65,66].

### 4.3. Locust Outbreaks in Croatia

#### 4.3.1. The Nature of the Croatian 2022 Locust Outbreaks

When the E/F ratios (1.66 and 1.69) of the gathered specimens of *Dociostaurus maroccanus* are compared to the values in Barranco [38], it can be seen that they correspond to the lower extreme of the range, describing the gregarious form. Similar values were obtained by Mikšić [51] in Herzegovina, who concluded that the observed specimens have not fully transitioned to the gregarious phase and in fact belong to the transient phase. Neither the outbreak in Štikovo nor the one reported by Mikšić [51] consisted of many migratory adults, which is contrasted by Gradojević [21,22], who reported flying swarms in the past. This prompts us to conclude that a true gregarious outbreak did not occur in Štikovo and that the gregarious individuals from Dalmatia and Herzegovina, if they are recorded in the future, will likely be characterized by higher E/F ratios than those reported herein.

The E/F ratio (1.27) of a single recorded *Calliptamus italicus* male from Brač is decidedly in the range of values describing the solitarious form [15]. A single individual does not represent an entire population, but considering that no flying swarms were reported and that the supposed outbreak quickly died down, it can be concluded that this, too, was not a true outbreak but probably a spike in population density below the gregarization threshold. A significant outbreak of *Calliptamus italicus* was recorded in Kuršumlija, Serbia, relatively recently [67], so significant outbreaks of Italian locusts should be monitored in the future.

It is important to note that the term “outbreak” officially describes populations that have undergone gregarization, although that terminology is not rigorously followed [68]. Throughout this paper, we refer to the herein-reported population spikes as “outbreaks” even though they do not fulfill the strict criteria for that designation. Although the recorded events are not nearly as severe as those that are usually reported [15,18], they represent an unusual occurrence for this area and are thus treated with greater diligence.

#### 4.3.2. Critical Segments of the Locusts’ Life Cycles

*Dociostaurus maroccanus* is a univoltine polyphagous species associated with steppe habitats that are most commonly populated by *Poa bulbosa* [69,70,71]. Its life cycle overlaps well with the life cycle of the grasses, with the nymphs hatching in spring and eating fresh plant matter, and adult females ovipositing during the summer among dry plants [69,72]. It is important to note that the eggs of this species can develop only in undisturbed compact soil. Developing a piece of land into a crop area makes it entirely unsuitable for the species [18]. Many people in Sardinia [19], just as in Dalmatinska Zagora [73], left the traditional way of life. Entire villages or parts of villages are now completely abandoned, so more and more areas are suitable for the development of the locust species, which could result in more outbreaks and thus threaten the remaining populace.

It is a well-established fact that the amount of rainfall during the spring is an essential factor that allows for the post-diapause development of Moroccan locust eggs. A total amount of about 100 mm of total precipitation during the spring (i.e., during March and April) seems to be the optimum for the species, while a rainier season could prove lethal for the eggs [69,72]. A look at the climatic data of the area reveals that it received about 130 mm of rainfall during the spring of 2022, while for the earlier years, this value equals approximately 90 mm (2021), 50 mm (2020), 100 mm (2019), 180 mm (2018), 130 mm (2017), 130 mm (2016), 66 mm (2015), and 200 mm (2014) [74]. The last 4 years have been drier than average, which supports the reports that the swarms have been appearing regularly during those 4 years. However, most of the other recorded years have not been prohibitively rainy, which means that the occurrence of a *D. maroccanus* outbreak cannot be easily predicted from single-year precipitation data. The lack of substantial reporting means that we lack critical data that would allow us to model shifts in the species’ population densities over longer periods of time. The areas that have had a low amount of precipitation through several consecutive years, and which are within the distribution area of this species, should be examined by experts in the future. It is critical to record rainfall as part of any effort to model outbreaks of locust species.

The Italian locust is a univoltine polyphagous species associated with steppe habitats rich in *Artemisia* spp. Its life cycle is similar to that of the Moroccan locust but with much less strict water preferences, i.e., the eggs can remain in diapause for a long time before rainfall comes [15,69]. It has been observed that the November and December rains kickstart the development of the eggs, and that there is an increased mortality of the eggs if the rains are late [69]. However, it appears that predicting the population densities of this species is not as simple as tracking rainfall during late fall. Research into the biology of the Italian locust revealed that many different factors are at play, and that even small changes in various environmental conditions can result in vastly different population densities [15]. As a general rule, dry years with rains in late fall should be an impetus to monitor the *C. italicus* populations more carefully, but ideally, local populations should be tracked with as many parameters as possible to allow for better predictions in the future [15,69]. The last two years have been exceptionally dry, with around half of the total rainfall happening in November and December. The earlier years had a more uniform rainfall regiment, with November and December getting around a quarter of the total rainfall [74]. This corroborates the idea that rainfall should be monitored as a warning of a potential upcoming outbreak. However, since Brač was not the only area with such a rainfall profile, it is obvious that many more factors are at play in determining the population density of *C. italicus*.

The spatiotemporally accurate reporting of outbreaks is important, so the information could be relevant for control planning and the allocation of control resources, but this requires the investment and training of a dedicated Orthoptera research unit in Croatia.

### 4.4. What Can and Should Be Done in the Future

The economic importance of *D. maroccanus* has decreased over time, with outbreaks happening less often [14,18], but Sardinia still remains a notable outbreak spot for this species [19]. The threat of *C. italicus* remains stable [15]. Despite the global trend, the climate of the Mediterranean region of Croatia and of Bosnia and Herzegovina remains suitable for both species. Furthermore, the breaking points of the two species’ life cycles are different but do not exclude each other, meaning that they can both form dense populations within the same year, as evidenced by the 2022 outbreaks. The trends of climate change in the Mediterranean region show increases in temperatures and decreases in annual rainfall, with most of it being concentrated in winter [14,75]. These are the exact conditions that favor the development of locusts [2,3], which implies that outbreaks might become more frequent in the region.

Our examination of the literature revealed five historical records of locust outbreaks in Croatia, all of them occurring in the Mediterranean part of the country. Some additional outbreaks were reported from the neighboring areas of Bosnia and Herzegovina. Some sources [21,22,27,51] treat locusts as known and unsurprising pests in the contexts of the areas where the outbreaks occurred, potentially signifying that these occurrences are more common than is reported.

*Locusta migratoria*, the Migratory locust, is distributed in the Mediterranean region of Croatia and is occasionally recorded in larger (but not destructive) numbers [48]. Historically, the species has been recorded as common in the delta of Neretva River [76], but no swarms have ever been recorded. The outbreaks of this species are a rare occurrence in the Balkans [77]. Recently, a large swarm of Migratory locusts was reported from Moldavia [78] and eastern Romania [78]; it happened in 2009, destroyed more than 50 hectares of corn crops just in Moldavia, and was contained by use of pesticides. An outbreak of Migratory locusts in Croatia seems unlikely but is certainly possible. At present, it is impossible to identify which combination of factors might cause gregarization, so this species should be monitored. Besides the aforementioned species, the locust species *Schistocerca gregaria*, without an established population in Europe, has been recorded in Croatia as well [48]. This species should be kept in mind as it could colonize Croatia in the future.

The response to the 2022 outbreaks was slow and muted. The authors were made aware of them through second- or third-hand accounts and were physically and financially unable to respond in time. There has been no damage assessment nor has any institution in Croatia issued guidelines for reporting and treating locust outbreaks. Since no damage reports have been produced, it is impossible to ascertain the true economic importance of outbreaks in Croatia. The only recorded information is that in 1946 and 1947 the Italian locust destroyed 600 hectares of cultivated land in Istria [27]. This implies that the threat to crops could be serious, especially considering that local population spikes of solitarious-form locusts could cause persistent damage that could add up over the entire region, with each of them not being deemed significant enough to report on its own. To test this, a large-scale research study on pest perception and presence in the Mediterranean part of Croatia should be conducted.

In areas where locust outbreaks are common and devastating, guidelines for reporting them have been produced [79]. In the simplest of terms, the following three main questions should be answered: (1) what was seen, (2) where was it seen, and (3) who saw it. Every report should contain as much information as possible, for example, which species was swarming, whether the adults or nymphs were observed, an approximation of the population density, the behavior of the swarm (mating or laying eggs), when it started, and how long it lasted. Ideally, the data should be accompanied by video and photographic footage. Such a set of guidelines should be produced and made available to the local populace, and the local experts should be called in to assess any future outbreak.

The media are prone to sensationalistic reporting and cannot be expected to provide data of the appropriate level of quality; the local institutions should be the ones to dedicate some resources to this task. However, during the last two years, the local media were the only ones reporting on the swarms, albeit in a fearmongering and information-poor way. There is a need to establish a direct communication channel between experts and the local populace in order to gather real data on Orthoptera mass occurrences in the country. The current way of gathering data is ineffective. The populace claims that outbreaks are regularly reported to authorities, but there are usually no follow-ups. As can be seen from the statements by the local institutes of public health [41], they do not distinguish caeliferan and ensiferan outbreaks, and their data are not publicly available despite containing information on some large outbreaks. It is essential to communicate to the general public the importance of recording these events and to whom they can report them. The Croatian government should give more support to scientific research on this matter and provide quick and realistic help to local communities that are suffering from or are at risk of outbreaks.

## Figures and Tables

**Figure 1 insects-15-00082-f001:**
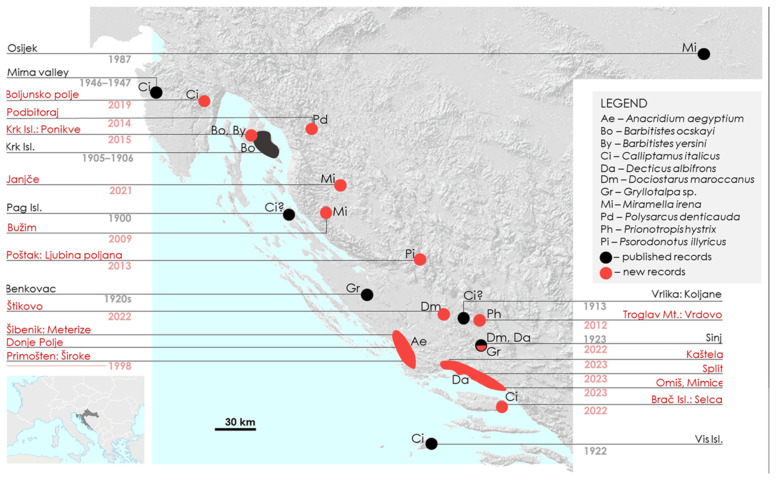
The map of Croatia with the locations of the outbreaks recorded in the country. The map shows locations of all the mass occurrences, the year of each occurrence, and the species. Red dots show previously unpublished records, while black ones point to the literature records.

**Figure 2 insects-15-00082-f002:**
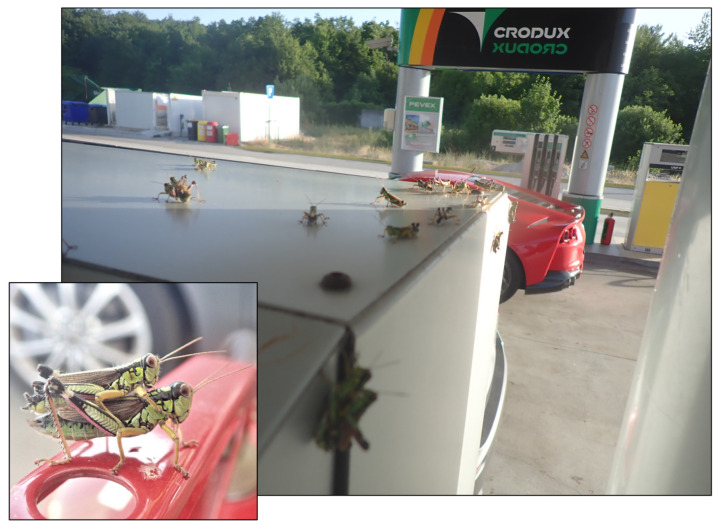
*Miramella irena* mass occurrence at Odmorište Janjče. Observed by N. Tvrtković and M. Vuković on 29 June 2021.

**Figure 3 insects-15-00082-f003:**
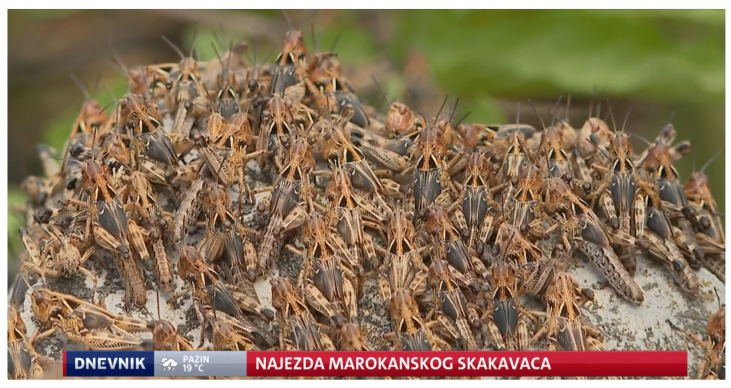
Moroccan locust nymphs. *Dociostaurus maroccanus* outbreak in Svilaja Mt., Štikovo from Dnevnik Nove TV News. Photo by Dnevnik Nove TV, reproduced with permission.

**Figure 4 insects-15-00082-f004:**
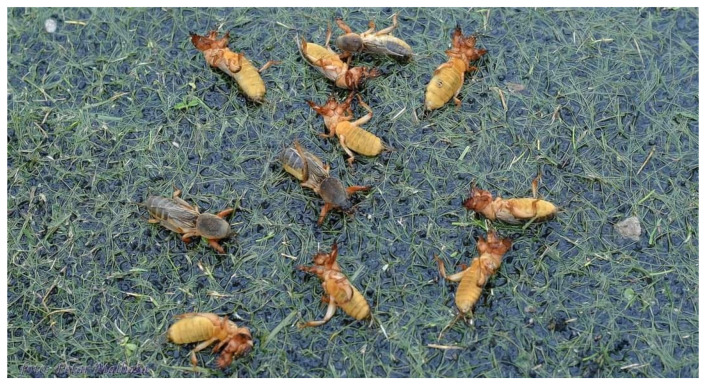
Adult brachypterous *Gryllotalpa* sp. in Sinj on NK Junak field. Photo by Petar Malbaša, published on Dalmatinski portal. Reproduced with permission.

**Table 1 insects-15-00082-t001:** List of all the localities mentioned in the study, with annotated biogeographical regions according to the Checklist of Croatian Orthoptera [48], together with the accompanying coordinates. It is not clear which section of forestry around Osijek Pavićević and Karaman [28] is being referred to, but Tikveš Forestry is the closest to Osijek city.

Locality	Biogeographical Region	Coordinates
Benkovac	Adriatic region: Dalmatia	44.03 N, 15.61 E
Boljunsko polje (near Kurelovići)	Adriatic region: Istria	45.29 N, 14.13 E
Bužim (Velebit Mt.)	Dinaric Alps region	44.58 N, 15.24 E
Brač Island: Selca	Adriatic region: islands	43.30 N, 16.84 E
Donje Polje by Šibenik	Adriatic region: Dalmatia	43.69 N, 15.96 E
Janjče gas station (A1 highway)	Dinaric Alps region	44.751 N, 15.373 E
Kaštela: Kaštel Štafilić	Adriatic region: Dalmatia	43.537 N, 16.299 E
Kaštela: Kaštel Stari	Adriatic region: Dalmatia	43.56 N, 16.345 E
Krk Island: Baška	Adriatic region: islands	44.97 N, 14.75 E
Krk Island: Draga Bašćanska	Adriatic region: islands	44.99 N, 14.72 E
Krk Island: Kornić	Adriatic region: islands	45.05 N, 14.61 E
Krk Island: Punat,	Adriatic region: islands	45.02 N, 14.63 E
Krk Island: Ponikve lake	Adriatic region: islands	45.07 N, 14.57 E
Krk Island: Vrbnik	Adriatic region: islands	45.07 N, 14.67 E
Mimice	Adriatic region: Dalmatia	43.411 N, 16.787 E
Mirna valley (Istria)	Adriatic region: Istria	45.35 N, 13.69 E
Osijek: Tikveš, Darda, or Valpovo Forestry	Pannonian region	(45.6 N, 18.7 E)
Pag Island	Adriatic region: islands	44.56 N, 14.90 E
Podbitoraj	Dinaric Alps region	45.11 N, 15.10 E
Poštak Mountain: Ljubina poljana	Dinaric Alps region	44.26 N, 16.10 E
Primošten: Široke	Adriatic region: Dalmatia	43.59 N, 16.01 E
Sinj	Adriatic region: Dalmatia	43.70 N, 16.65 E
Split: Mejaši	Adriatic region: Dalmatia	43.52 N,16.48 E
Svilaja Mountain: Štikovo	Adriatic region: Dalmatia	43.91 N, 16.31 E
Šibenik: Meterize	Adriatic region: Dalmatia	43.75 N, 15.89 E
Troglav Mt.: Vrdovo	Adriatic region: Dalmatia	43.86 N, 16.64 E
Vis Island	Adriatic region: islands	43.04 N, 16.20 E
Vrlika: Koljane	Adriatic region: Dalmatia	43.88N, 16.50E

**Table 2 insects-15-00082-t002:** Checklist of the 11 Orthoptera species of known identity (Gryllidae sp. reported by Langhoffer 1927 excluded) with known mass occurrence in Croatia, taxonomically sorted by suborder, family, and subfamily, with the English and Croatian vernacular names given for each species.

Taxonomic Checklist	English Vernacular Name	Croatian Vernacular Name
Suborder Ensifera		
Family Tettigoniidae		
Subfamily Phaneropterinae		
*Barbitistes ocskayi* Charpentier in Ocskay et al., 1850	Black Saw Bush-cricket	Ocskayjev ljuskokrili konjic, Ocskayeva kršuljka
*Barbitistes yersini* Brunner von Wattenwyl, 1878	Balkan Saw Bush-cricket	Yersinov ljuskokrili konjic, Yersinova kršuljka
*Polysarcus denticauda* (Charpentier, 1825)	Bull Bush-cricket	mesnati bodljorepi konjic
Subfamily Tettigoniinae		
*Decticus albifrons* (Fabricius, 1775)	Mediterranean Wart-biter	veliki primorski konjic
*Psorodonotus illyricus* Ebner, 1923	Illyrian Walking Bush-cricket	ilirski oklopljeni konjic
Family Gryllotalpidae		
Subfamily Gryllotalpinae		
*Gryllotalpa* sp.	Mole-cricket	rovac
Suborder Caelifera		
Family Acrididae		
Subfamily Cyrtacanthacridinae		
*Anacridium aegyptium* (Linnaeus, 1764)	Egyptian Bird Grasshopper	egipatska šaška
Subfamily Calliptaminae		
*Calliptamus italicus* (Linnaeus, 1758)	Common Pincer Grasshopper	talijaski krupnozadi skakavac
Subfamily Melanoplinae		
*Miramella irena* (Fruhstorfer, 1921)	Long-winged Mountain Grasshopper	lijepa irena
Subfamily Gomphocerinae		
*Dociostaurus maroccanus* (Thunberg, 1815)	Moroccan Cross-backed Grasshopper	crvenonogi X-skakavac
Family Pamphagidae	Toad Grasshoppers	žaboliki skakavci
Subfamily Thrinchinae		
*Prionotropis hystrix* (Germar, 1817)	Eastern Stone Grasshopper	krški žaboliki skakavac

**Table 3 insects-15-00082-t003:** Checklist of the 11 Orthoptera species of known identity (Gryllidae sp. reported by Langhoffer [25] excluded) with known mass occurrence in Croatia, taxonomically sorted by suborder, family, and subfamily, with the English and Croatian vernacular names given for each species. Question marks (?) represent questionable identification.

Year	Species	Locality	Source of Data
1900	? (*Calliptamus italicus*)	Pag Island	[26]
1905	*Barbitistes ocskayi*	Krk island: Draga Bašćanska	[24,25]
1906	*Barbitistes ocskayi*	Krk island: Punat, Vrbik, Kornić	[24,25]
1920s	? (*Gryllus* sp./Gryllidae sp.)	Krk island: Baška	[25]
1920s	*Gryllotalpa* sp.	Benkovac	[25]
1913	? (*Calliptamus italicus*)	Vrlika: Koljane	[26]
1922	*Calliptamus italicus*	Vis Island	[26]
1923	*Dociostaurus maroccanus*	Sinj	[26]
1923	*Decticus albifrons*	Sinj	[26]
1946/47	*Calliptamus italicus*	Istra: Mirna valley	[27]
1987	*Miramella irena*	Forestry near Osijek	[28]
1998	*Anacridium aegyptium*	From Šibenik (Meterize) via Donje Polje to Primošten (Široke)	This study
2009	*Miramella irena*	Velebit Mt.: Bužim	This study (obs. on 8.7. by N. Tvrtković, M. Vuković, I. Mihoci)
2012	*Prionotropis hystrix*	Troglav Mt.: Vrdovo	This study (obs. on 3.8. by L. Ivić)
2013	*Psorodonotus illyricus*	Poštak Mt.: Ljubina Poljana	This study (obs. on 17.6. by N. Tvrtković, M. Vuković)
2014	*Polysarcus denticauda*	Podbitoraj	This study (obs.in August by N. Tvrtković)
2015	*Barbitistes ocskayi, B. yersini*	Krk Island	This study (obs. C. Roesti, J. Skejo)
2019	*Calliptamus italicus*	Boljunsko polje	This study (obs. N. Tvrtković)
2021	*Miramella irena*	Janjče	This study (obs. on 29.6. by N. Tvrtković, M. Vuković,)
2022	*Dociostaurus maroccanus*	Svilaja: Štikovo	This study
2022	*Calliptamus italicus*	Brač island	This study
2022	*Gryllotalpa* sp.	Sinj	This study
2023	*Decticus albifrons*	Kaštela, Split, Sinj	This study

## Data Availability

The original contributions presented in the study are included in the article. Further inquiries can be directed to the corresponding authors.

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
