# Peer review of "An Overview of Orthoptera Mass Occurrences in Croatia from 1900 to 2023"

_insects, 2024, doi:10.3390/insects15020082_

Round 1

Reviewer 1 Report

Comments and Suggestions for Authors

The authors give an overview on what is known on mass occurrences of Orthoptera species in Croatia from 1900 to 2023 based on literature studies and for more recent times own observations and research. Although several species were recorded to occur in high numbers   in one or a few years during the time considered for the study, only three species of Caelifera were identified as pest species Dociostaurus maroccanus, Anacridium aegyptium, Calliptamus italicus. The status of the "Croatian locust" was not clearly defined.

The manuscript is written in good and understandable English, the progression of the presentation of data is consistent and the conclusions drawn from the results are logical. There are only a few points to be considered for correction:

Lines: 

106: ... who happened to be on the mountain at the moment. – ... who happened to visit the mountain during the outbreak.

181–182: ... of which 11 have their identity known. – ... of which 11 are known to species level.

427–428: ... but no broader conclusions present themselves at this moment. – ... but we to not have any conclusions why it is like this.

475–476: ... about 130 mm of rainfall during the spring of 2022, while for the earlier years, this value equals approximately 90, 50, 100, 180, 130, 130, 66, and 200 ... 

Remark: How can you conclude from this sequence that "It is evident that the last 4 years have been drier than average" (line 477) ? It appears to be opposite.

Comments on the Quality of English Language

Generally of good quality. A few minor points are marked  in the manuscript.

Author Response

The authors give an overview on what is known on mass occurrences of Orthoptera species in Croatia from 1900 to 2023 based on literature studies and for more recent times own observations and research. Although several species were recorded to occur in high numbers in one or a few years during the time considered for the study, only three species of Caelifera were identified as pest species Dociostaurus maroccanus, Anacridium aegyptium, Calliptamus italicus.

Reply: Thank you very much for your review.

The status of the "Croatian locust" was not clearly defined.

Reply: Thank you for the comment. We have removed the phrase.

The manuscript is written in good and understandable English, the progression of the presentation of data is consistent and the conclusions drawn from the results are logical. There are only a few points to be considered for correction:

Reply: Thank you very much, please find each point answered.

Lines: 

106: ... who happened to be on the mountain at the moment. – ... who happened to visit the mountain during the outbreak.

Reply: Thank you. Corrected.

181–182: ... of which 11 have their identity known. – ... of which 11 are known to species level.

Reply: Thank you. Corrected.

427–428: ... but no broader conclusions present themselves at this moment. – ... but we to not have any conclusions why it is like this.

Reply: Thank you. Corrected to „we do not have any conclusions why it could be like this.“

475–476: ... about 130 mm of rainfall during the spring of 2022, while for the earlier years, this value equals approximately 90, 50, 100, 180, 130, 130, 66, and 200 ...  Remark: How can you conclude from this sequence that "It is evident that the last 4 years have been drier than average" (line 477) ? It appears to be opposite.

Reply: Thank you for the comment. We made it clear by putting the year next to the value for each year, „90 mm (2021), 50 mm (2020), 100 mm (2019), 180 mm (2018), 130 mm (2017), 130 mm (2016), 66 mm (2015), and 200 mm (2014)“, now it is visible that from 2019 to 2023, rainfall was much lower than before (130, 90, 50, 100 vs. 180, 130, 130, 66, 200).

Reviewer 2 Report

Comments and Suggestions for Authors

Reviewer’s comments:

Page 1, lines 40-41

Printed:

Keywords: swarm, najezda, najezda skakavaca massive occurrence, Dociostaurus maroccanus, Anacridium aegyptium, Calliptamus italicus, Croatian locust outbreaks, pest species

Replace to:

Keywords: Mediterranean region, locusts, grasshoppers, katydids, mole crickets, pest species, abundance, outbreaks

Page 3, line 137

Printed:

... interviews with local populace were conducted in order to ....

Replace to:

... interviews with local peoples were conducted in order to ....

Page 5, line 182

Printed:

... while for the 1920s Krk outbreak, it is not known ...

Replace to:

... while for the 1920s Krk island outbreak, it is not known ...

Page 5, Table 2

At and of table line

Taxonomic checklist

English vernacular name

Croatian vernacular name

must be deleted!

Page 8, line 271

Printed:

... a 25-year-old Šibenik Anacridium aegyptium swarm ...

Replace to:

... a 25-year-old Šibenik Anacridium aegyptium swarm ...

Page 9, lines 328-329

Printed:

On 7 and 8 June 2015, a mass appearance of Barbitistes ocskayi and Barbitistes yersini was observed ...

Replace to:

On 7 and 8 June 2015, a mass appearance of Barbitistes ocskayi and B. yersini, 1878 was observed ...

Page 10, line 361

How many species of the genus Gryllotalpa are recorded from Croatia? The specimens in Fig. 4 look very similar to brachypterous form of Gryllotalpa gryllotalpa widely distributed in Europe. Why you use in text Gryllotalpa sp.? According to Harz (1969: 737) namely G. gryllotalpa has dark brown transversal vienlets of the tegmina (like in Fig 3), while in G. unispina such veinlets light (yellowish).

Author Response

Page 1, lines 40-41 Printed: Keywords: swarm, najezda, najezda skakavaca massive occurrence, Dociostaurus maroccanusAnacridium aegyptium, Calliptamus italicus, Croatian locust outbreaks, pest species

Replace to: Keywords: Mediterranean region, locusts, grasshoppers, katydids, mole crickets, pest species, abundance, outbreaks

Reply: Thank you for the comment. We have replaced the keywords as suggested, but next to the word outbreak we kept in the brackets “outbreaks (hrv. najezda)“, so Croatian speakers can easily find the paper when searching for „najezda“.

Page 3, line 137 Printed: ... interviews with local populace were conducted in order to ....

Replace to: ... interviews with local peoples were conducted in order to ....

Reply: Thank you, corrected to „interviews with local people”

Page 5, line 182 Printed: ... while for the 1920s Krk outbreak, it is not known ...

Replace to: ... while for the 1920s Krk island outbreak, it is not known ...

Reply: Thank, you, the word “island” is added.

Page 5, Table 2

At and of table line “taxonomic checklist, English vernacular name, Croatian vernacular name” must be deleted!

Reply: Thank you, corrected.

Page 8, line 271 Printed: ...

a 25-year-old Šibenik Anacridium aegyptium swarm ...

Replace to: ... a 25-year-old Šibenik Anacridium aegyptium swarm ...

Reply: Thank you, but in our version of Anacridium aegyptium in already in italic.

Page 9, lines 328-329 Printed: On 7 and 8 June 2015, a mass appearance of Barbitistes ocskayi and Barbitistes yersini was observed ...

Replace to: On 7 and 8 June 2015, a mass appearance of Barbitistes ocskayi and B. yersini, 1878 was observed ...

Reply: Thank you, corrected, and furthermore, number “1878” was removed.

Page 10, line 361

How many species of the genus Gryllotalpa are recorded from Croatia? The specimens in Fig. 4 look very similar to brachypterous form of Gryllotalpa gryllotalpa widely distributed in Europe. Why you use in text Gryllotalpa sp.? According to Harz (1969: 737) namely G. gryllotalpa has dark brown transversal vienlets of the tegmina (like in Fig 3), while in G. unispina such veinlets light (yellowish).

Reply: Thank you for your comment. When Harz (1969) wrote Die Orthpteren Europas, die situation with Gryllotalpa was much less complex than today. For now, there are at least 3 Gryllotalpa species in Croatia (see Skejo et al. (2018). The first annotated checklist of Croatian crickets and grasshoppers (Orthoptera: Ensifera, Caelifera). Zootaxa, 4533(1), 1-95.). Presence of G. krimbasi is likely (see Iorgu et al. (2016). Geographic distribution of Gryllotalpa stepposa in south-eastern Europe, with first records for Romania, Hungary, and Serbia (Insecta, Orthoptera, Gryllotalpidae). ZooKeys, (605), 73.), but the main problem is that on the other side of the Adriatic sea, there are 5+ Gryllotalpa species with dark tegminal veins defined by the number of chromosomes (see Massa et al. (2012) Fauna d'Italia, Volume 48: Orthoptera), and since our coast was connected to Italian only 12 000 years ago, there is chance that Croatia has more than 3 Gryllotalpa species. Mediterranean region is rich in brachypterous forms, and for these review is necessary in order so assign them with certainty to any species.

Reviewer 3 Report

Comments and Suggestions for Authors

Title: An overview of Orthoptera mass occurrences in Croatia from 1900 to 2023

I presume that this is a review article?

An interesting review/synopsis where the authors have obviously spent a lot of time and energy researching various historical and contemporary outbreaks of locusts and grasshopper species in Croatia. Unfortunately, there is almost no scientific content to back up the various anecdotal reports or witness accounts of outbreaks of the various Orthoptera species. However, the manuscript is still a valuable historical synopsis of outbreaks in Croatia that will hopefully form a baseline for more scientific data reports in future. A lot of editorial assistance is still required throughout much of the manuscript.

Line 30. Destructivity? Unusual use of language? Suggest edit.

A total of 23 mass outbreaks of Orthoptera species from 1900 to 2023, with 28 localities and 12 species. Interesting historical records for Croatia, but this is a low level of outbreak activity compared with other locust and pest grasshopper species in other parts of the world.

Reported 6 outbreaks of Calliptamus italicus, 3 outbreaks of 2 x grasshoppers and 2 outbreaks of Dociostaurus maroccanus. Also 1 x outbreak of Anacridium aegyptium in 1998 (not given genus name here at first citing!). Recent 2022 outbreak of D. maroccanus reported, which is very interesting.

Introduction

Line 44 to L60. Provides a basic and rather simplistic overview of the well known and often recited definition of locusts and the density dependent phase change phenomenon. Suggest rewrite and shorten.

Line 58 and L59.  Terms such as ‘accompanying voracity’ and ’annihilate crops’ are too dramatic and are not correct for a scientific paper. Revise.

 Line 61 and L62. No mention of the book by Uvarov (1977), who gave a detailed review of the distribution and ecology and outbreak dynamics of the Moroccan locust, page 460 to 476. Please refer to this important source. 

Line 68. Use of the sentence ‘plagued by locust swarms for hundreds of years’ is considered by the reviewer as an over exaggeration of the case when only 23 outbreaks of Orthoptera have been reported in the past 123 years.

Line 78. No mention of literature of the publications by Adamovic, Z.R, (1956; 1959; 1964; 1968) cited in Uvarov (1977) regarding the occurrence of the Italian locust and the Moroccan Locust, as well as the Migratory locust and grasshopper outbreaks in various areas of the previous Yugoslavia (e.g. Banat area in Serbia, Ulcinj District in Montenegro). Suggest investigate if these references are relevant for the current manuscript?.

Line 83 to 84. Reviewer considers that the authors undertook a valuable exercise to summarize all existing 11 literature records and to report on 12 undocumented outbreaks. 

Materials and Methods

Line 94. Literature review going back to 1907

Literature survey and collation of anecdotal records, newspaper reports, some photographs and witness accounts and interviews conducted by authors. Authors certainly tried hard to obtain as much information as possible, but their information resources were sparse.The authors have therefore done a good job of investigative reporting.

 Line 97 to L.101 were author’s direct observations, which are considered valid.

Line 101 to L.144. Anecdotal observations and reports by various people, but no reason to doubt their actual observation even though no scientific data are available to support the reports. 

Results

Line 156. There are 23 records of outbreaks in Croatia. 11 records from the literature and 12 new records.

Line 166 to L170. Sinj is city with highest frequency of outbreaks, with 3 records in past 100 years. This is low frequency and novelty value only. Krk island had most localized outbreaks with 6 of the 28 localities, but most of the records date back to 1905! Noted that other outbreaks could have occurred but were not reported.

Line 172 to L.180. A total of 5 Orthoptera outbreaks caused damage in Croatia in period 1905 to 2022.  Reports relate to 12 species. This is a very low incidence of outbreaks, but is still considered to be interesting and valuable information for Croatia.

Line 242. Mass occurrences in Croatia.

Detailed detective work undertaken by authors regarding the origin and location of outbreaks.

Line 271. Reconstructing a 25-year-old Šibenik Anacridium aegyptium swarm from multiple personal accounts.

Again there was good detective work done here to elucidate this outbreak. However, these were very anecdotal reports and reminisces of the outbreak. Interesting records of a grasshopper outbreak (one or more species?), but there is absolutely no scientific observations to support the reports. I do agree with the authors that this is likely to be A. aeagytium as this is a large and unmistakable grasshopper species.  Paragraph needs editorial assistance.  

Line 307. Organized orthopteran research started in Croatia in 2010, so records post this date are more systematic and scientific, with 11 of the 23 records occurring after the formation of the research effort.

Line 307 to L.331. List of more recent grasshopper outbreaks, but no data given at all on areas infested, densities of grasshoppers causing damage, instars observed, movement of grasshoppers, etc.  This was a missed opportunity to collect scientific data.

Line 332 to L.342. Recent swarm outbreak of Dociostaurus maroccanus in Štikovo. More data given here, but only 2 x males available for E/F ratio measurements and no damage assessments made. Unfortunately this was a missed opportunity to collect data on a rare outbreak event.

Line 343 to L.347. Report of Moroccan locust outbreak on Brač island is very interesting, but again this was a missed opportunity to collect more data.

Line 351 to L.360. C. italicus on Brač report. Unfortunately, there is no supporting data.   

 Line 370 to L.373. In 2023 there was a massive upsurge in the population of Decticus albifrons in the Dalmatian region, most notably in Kaštela, Split, and Omiš. Very interesting and valuable record on an ensiferan outbreak.

Discussion

Line 376 to L.396. The authors now start talking about Ensiferans which is a distraction when there is little data to share.

Line 415 to L.429. More discussion on Anacriduium aegyptium outbreak here. A lot was said about this outbreak earlier in the text, so some of the information is being repeated. 

Line 432 to L.450. A discussion is being held based on very little field evidence of E/F ratios.

Line 451 to L.457. I agree with authors that it is valid to describe unusually high population densities as grasshopper ‘outbreaks’ even if there is no phase change involved.

Line 459 to L.469. Life cycle of Moroccan locust should have perhaps been given earlier in text when first describing outbreaks in Croatia.

Line 470 to L.485. Interesting observations on rainfall and local outbreaks of the Moroccan locust. Conclusions are a bit weak based on little information being presented, but worth investigating further.  

Line 502 to L.503. It is certainly critical to record rainfall as part of any effort to model outbreaks of these locust species.

Line 560 to L.574. The authors discuss the importance of accurate and timeous reporting of outbreaks, so that the information is relevant for control planning and allocation of control resources. This requires investment and training of a dedicated Orthoptera unit in Croatia.

Comments on the Quality of English Language

Quality of the English used in the text is poor in parts. Manuscript requires editorial assistance to remove the more dramatic and 'flowery' statements. Text needs to be condensed and duplication avoided.

Author Response

Title: An overview of Orthoptera mass occurrences in Croatia from 1900 to 2023

I presume that this is a review article? An interesting review/synopsis where the authors have obviously spent a lot of time and energy researching various historical and contemporary outbreaks of locusts and grasshopper species in Croatia. Unfortunately, there is almost no scientific content to back up the various anecdotal reports or witness accounts of outbreaks of the various Orthoptera species. However, the manuscript is still a valuable historical synopsis of outbreaks in Croatia that will hopefully form a baseline for more scientific data reports in future. A lot of editorial assistance is still required throughout much of the manuscript.

Reply: Thank you very much for your words and your review. Indeed, this is a review article aiming to show, among other things, how bad and inaccurate was the swarm detection and documentation in the past. Your review provided many useful comments and suggestions, for which we thank you once again. Please find response to each of your points below.

Line 30. Destructivity? Unusual use of language? Suggest edit.

Reply: Thank you for the comment. We have removed the word from the text.

A total of 23 mass outbreaks of Orthoptera species from 1900 to 2023, with 28 localities and 12 species. Interesting historical records for Croatia, but this is a low level of outbreak activity compared with other locust and pest grasshopper species in other parts of the world.

Reply: Thank you, we agree. We have added a sentence to the abstract “This is a low level of outbreak activity compared with other locust and pest grasshopper species in other parts of the world.”

Reported 6 outbreaks of Calliptamus italicus, 3 outbreaks of 2 x grasshoppers and 2 outbreaks of Dociostaurus maroccanus. Also 1 x outbreak of Anacridium aegyptium in 1998 (not given genus name here at first citing!). Recent 2022 outbreak of D. maroccanus reported, which is very interesting.

Reply: Thank you, we have added full genus name of Anacridium.

Line 44 to L60. Provides a basic and rather simplistic overview of the well known and often recited definition of locusts and the density dependent phase change phenomenon. Suggest rewrite and shorten.

Reply: Thank you for your comment, but this already is an abbreviated version with basic info only.

Line 58 and L59.  Terms such as ‘accompanying voracity’ and ’annihilate crops’ are too dramatic and are not correct for a scientific paper. Revise.

Reply: Thank you. We have replaced “annihilate” with “destroy”.

 Line 61 and L62. No mention of the book by Uvarov (1977), who gave a detailed review of the distribution and ecology and outbreak dynamics of the Moroccan locust, page 460 to 476. Please refer to this important source. 

Reply: Thank you, we have included Uvarov (1977).

Line 68. Use of the sentence ‘plagued by locust swarms for hundreds of years’ is considered by the reviewer as an over exaggeration of the case when only 23 outbreaks of Orthoptera have been reported in the past 123 years.

Reply: Thank you for your comment, but our sentence refers to the whole Balkans, including Greece, Albania, Northern Macedonia, Bulgaria, southern Romania, Serbia, Montenegro, Bosnia and Herzegowina and these areas were indeed plagued by locust swarms for hundreds of years. For Croatia, there are fewer data than for the other countries, but it is part of the Balkans.

Line 78. No mention of literature of the publications by Adamovic, Z.R, (1956; 1959; 1964; 1968) cited in Uvarov (1977) regarding the occurrence of the Italian locust and the Moroccan Locust, as well as the Migratory locust and grasshopper outbreaks in various areas of the previous Yugoslavia (e.g. Banat area in Serbia, Ulcinj District in Montenegro). Suggest investigate if these references are relevant for the current manuscript?.

Reply: Thank you for your comment. We have these papers, but they are not relevant to our study. There are many reported outbreaks in the neighboring countries, but it was not the topic of our study.

Line 83 to 84. Reviewer considers that the authors undertook a valuable exercise to summarize all existing 11 literature records and to report on 12 undocumented outbreaks. 

Reply: Thank you very much.

Line 94. Literature review going back to 1907

Literature survey and collation of anecdotal records, newspaper reports, some photographs and witness accounts and interviews conducted by authors. Authors certainly tried hard to obtain as much information as possible, but their information resources were sparse.The authors have therefore done a good job of investigative reporting.

Reply: Thank you very much for your nice words.

Line 97 to L.101 were author’s direct observations, which are considered valid.

Reply: Yes, thank you.

Line 101 to L.144. Anecdotal observations and reports by various people, but no reason to doubt their actual observation even though no scientific data are available to support the reports. 

Reply: Thank you, we agree.

Line 156. There are 23 records of outbreaks in Croatia. 11 records from the literature and 12 new records.

Reply: Yes, correct.

Line 166 to L170. Sinj is city with highest frequency of outbreaks, with 3 records in past 100 years. This is low frequency and novelty value only. Krk island had most localized outbreaks with 6 of the 28 localities, but most of the records date back to 1905! Noted that other outbreaks could have occurred but were not reported.

Reply: Thank you. We have added a sentence “however, this is a low frequency and likely of novelty value only.” And “It should be noted that other outbreaks could have occurred, but were not reported.”

Line 172 to L.180. A total of 5 Orthoptera outbreaks caused damage in Croatia in period 1905 to 2022.  Reports relate to 12 species. This is a very low incidence of outbreaks, but is still considered to be interesting and valuable information for Croatia.

Reply: Thank you, we agree.

Line 242. Mass occurrences in Croatia.

Detailed detective work undertaken by authors regarding the origin and location of outbreaks.

Reply: Thank you.

Line 271. Reconstructing a 25-year-old Šibenik Anacridium aegyptium swarm from multiple personal accounts.

Again there was good detective work done here to elucidate this outbreak. However, these were very anecdotal reports and reminisces of the outbreak. Interesting records of a grasshopper outbreak (one or more species?), but there is absolutely no scientific observations to support the reports. I do agree with the authors that this is likely to be A. aeagytium as this is a large and unmistakable grasshopper species.  Paragraph needs editorial assistance.  

Reply: Thank you, we are going to consult the editors and if needed request editorial assistance.

Line 307. Organized orthopteran research started in Croatia in 2010, so records post this date are more systematic and scientific, with 11 of the 23 records occurring after the formation of the research effort.

Reply: Yes, correct.

Line 307 to L.331. List of more recent grasshopper outbreaks, but no data given at all on areas infested, densities of grasshoppers causing damage, instars observed, movement of grasshoppers, etc.  This was a missed opportunity to collect scientific data.

Reply: Yes, correct. We agree and hope that in the future the data will be collected with more accuracy.

Line 332 to L.342. Recent swarm outbreak of Dociostaurus maroccanus in Štikovo. More data given here, but only 2 x males available for E/F ratio measurements and no damage assessments made. Unfortunately this was a missed opportunity to collect data on a rare outbreak event.

Reply: Yes, correct. We agree and hope that in the future the data will be collected more carefully and on time.

Line 343 to L.347. Report of Moroccan locust outbreak on Brač island is very interesting, but again this was a missed opportunity to collect more data.

Reply: Yes, correct. We agree.

Line 351 to L.360. C. italicus on Brač report. Unfortunately, there is no supporting data.   

Reply: Yes, correct. We agree. It was a missed opportunity.

Line 370 to L.373. In 2023 there was a massive upsurge in the population of Decticus albifrons in the Dalmatian region, most notably in Kaštela, Split, and Omiš. Very interesting and valuable record on an ensiferan outbreak.

Reply: Yes, thank you.

Line 376 to L.396. The authors now start talking about Ensiferans which is a distraction when there is little data to share.

Reply: We know there are only a few data on Ensifera, but we regarded they are important to be presented.

Line 415 to L.429. More discussion on Anacriduium aegyptium outbreak here. A lot was said about this outbreak earlier in the text, so some of the information is being repeated. 

Reply: Thank you, but we think some new information is discussed here in context.

Line 432 to L.450. A discussion is being held based on very little field evidence of E/F ratios.

Reply: Thank you. We agree it is little evidence, but unfortunately small sample was avalibale to us only.

Line 451 to L.457. I agree with authors that it is valid to describe unusually high population densities as grasshopper ‘outbreaks’ even if there is no phase change involved.

Reply: Thank you.

Line 459 to L.469. Life cycle of Moroccan locust should have perhaps been given earlier in text when first describing outbreaks in Croatia.

Reply: Earlier, it was part of the results, so we thought it better fits the Discussion section.

Line 470 to L.485. Interesting observations on rainfall and local outbreaks of the Moroccan locust. Conclusions are a bit weak based on little information being presented, but worth investigating further.  

Reply: Thank you, we agree.

Line 502 to L.503. It is certainly critical to record rainfall as part of any effort to model outbreaks of these locust species.

Reply: Thank you for your comment. We have added the sentence “It is critical to record rainfall as part of any effort to model outbreaks of the locust species.”

Line 560 to L.574. The authors discuss the importance of accurate and timeous reporting of outbreaks, so that the information is relevant for control planning and allocation of control resources.

This requires investment and training of a dedicated Orthoptera unit in Croatia.

Reply: Thank you for your comment. We have added the sentence to the end of this paragraph “Spatiotemporally accurate reporting of outbreaks is important, so the information for control planning and allocation of control resources can be relevant, but this requires the investment and training of a dedicated Orthoptera research unit in Croatia.”